# Precision Medicine Approaches in Acute Myeloid Leukemia with Adverse Genetics

**DOI:** 10.3390/ijms25084259

**Published:** 2024-04-11

**Authors:** Nicole Santoro, Prassede Salutari, Mauro Di Ianni, Andrea Marra

**Affiliations:** 1Hematology Unit, Department of Hematology and Oncology, Ospedale Civile “Santo Spirito”, 65122 Pescara, Italy; prassede.salutari@asl.pe.it (P.S.); mauro.diianni@unich.it (M.D.I.); 2Department of Medicine and Science of Aging, “G.D’Annunzio” University of Chieti-Pescara, 66100 Chieti, Italy; 3Laboratory of Molecular Medicine and Biotechnology, Department of Medicine, University Campus Bio-Medico of Rome, 00128 Rome, Italy; 4Institute of Translational Pharmacology, National Research Council of Italy (CNR), 00196 Rome, Italy

**Keywords:** acute myeloid leukemia, adverse genetics, target-immunotherapy

## Abstract

The treatment of acute myeloid leukemia (AML) with adverse genetics remains unsatisfactory, with very low response rates to standard chemotherapy and shorter durations of remission commonly observed in these patients. The complex biology of AML with adverse genetics is continuously evolving. Herein, we discuss recent advances in the field focusing on the contribution of molecular drivers of leukemia biogenesis and evolution and on the alterations of the immune system that can be exploited with immune-based therapeutic strategies. We focus on the biological rationales for combining targeted therapy and immunotherapy, which are currently being investigated in ongoing trials, and could hopefully ameliorate the poor outcomes of patients affected by AML with adverse genetics.

## 1. Background

The outcomes for patients with AML with adverse genetics remain poor, with a median overall survival (OS) of less than one year [1,2]. Adverse-risk or high-risk (HR) genetic AML encompasses several genetically defined entities representing about 50% of all adult AML cases [3]. HR-AML is more commonly characterized by a poor response to standard chemotherapy, a very short period of remission, and an increased rate of relapse even after allogeneic stem cell transplantation (allo-HCT). 

Novel compounds more recently introduced in the clinic, such as FLT3 or BCL2 inhibitors, have only demonstrated a modest impact on disease course in this specific AML category [4]. Currently, allo-HCT represents the sole potentially curative strategy for these patients, though survival rates rarely exceed 30–35% [5,6,7].

In this review, we present the latest advancements in the research field of HR-AML biology. We integrate insights from genomic analyses and from studies investigating the contribution of the immune system to myeloid leukemogenesis. Furthermore, we discuss the biological rationales behind the strategy of combining small molecules, which target specific genetic lesion(s), with immunotherapy. These combined treatment approaches are currently being investigated in ongoing clinical trials, holding promise for improving outcomes for patients with HR-AML.

## 2. Genetics of HR AML

HR AML represents an extremely complex subgroup of adult AML, characterized by a variety of well-defined cytogenetic and/or genetic lesions, which contribute to the aggressive course of the disease and its intrinsic resistance to standard chemotherapeutic approaches. In this section, we provide an overview of the current biological knowledge for each specific genetic entity of HR-AML, according to the European LeukemiaNet classification [1] (Table 1).

**t(6;9)(p23.3;q34.1)DEK::NUP214:** NUP214 is a nucleoporin that binds to the cytoplasmic side of the nuclear pore complex (NPC), which is critical for the nucleo-cytoplasmic transport of proteins and mRNA. Defective nuclear export derived from *DEK-NUP214* fusion induces the nuclear retention of transcription factors (TFs) that induce sustained *HOX* gene upregulation [2]. DEK is a chromatin-associated protein critical for the maintenance of chromatin stability. 

**t(v;11q23.3)*KMT2A*-rearranged.** Acute leukemias carrying *KMT2A* (MLL) translocations represent 5–10% of acute leukemia in all ages, and up to 70% of infantile leukemia [8]. *KMT2A* fusion supports leukemogenesis by recruiting the superelongation complex (SEC), the histone H3K79 methyltransferase DOT1L and menin (MEN1), to induce the overexpression of AML TFs such as HOXA9, MEIS1, and MEF2C [9]. *KMT2A*-rearranged leukemias feature promiscuous expression of lineage markers and a propensity for lineage switching [10,11].

**t(9;22)(q34.1;q11.2)*BCR::ABL1* (BCR-ABL+)**: This category comprises a subset of de novo AML developed in patients without a history of chronic myeloid leukemia (CML) and lacking recurrent genetic aberrations affecting the CEBPA or NPM1 gene, or cytogenetic alterations such as inv(16) or inv(3). Distinguishing BCR-ABL+ AML from a myeloid blast crisis of CML poses challenges. Unique to BCR-ABL+ AML are the loss of IKZF1 and CDKN2A, along with cryptic deletions in the IGH and TRG genes, features not observed in the myeloid blast crisis of CML [12]. AML blasts in this category often aberrantly express CD19, CD7, and TdT. Although BCR-ABL+ AML generally falls under the adverse-risk category, it should be noted that cases associated with inv(16) or *NPM1* mutations may have favorable outcomes [13,14,15].

**t(8;16)(p11.2;p13.3)*KAT6A::CREBBP***: This is a rare subset, representing 0.2 to 0.4% of all AML cases. CREBBP alterations in de novo AML have been reported to be associated with poor prognosis [16]. KAT6A, also known as MOZ or MYST3, encodes the monocytic leukemia zinc finger protein, a histone acetyltransferase of the MYST family that regulates gene transcription by activating the RUNX1 transcription factor complex. CREBBP plays a critical role in transcription regulation. Similar to KAT6A, CREBBP has an intrinsic histone acetyltransferase activity.

***EVI1-*rearranged:** *GATA2*, *MECOM(EVI1)* AML is characterized by the repositioning of a distal GATA2 enhancer that activates *MECOM* expression leading to GATA2 haploinsufficiency. About 20% of AML with inv(3)/t(3;3) harbor mutations in RUNX1, while around 25% exhibit mutations in IKZF1. Additionally, a subset of these AML cases presents with activating mutations in the RAS GTPase family member (NRAS or KRAS) or other signaling pathway proteins, such as PTPN11 and NF1, contributing to RAS signaling dysregulation and promoting AML cell proliferation. About 20% of patients have mutations in the polycomb protein ASXL1, and 30–60% have mutations in the spliceosomal machinery components, such as SF3B1 and U2AF1. *TP53* mutations are found in approximately 25% of cases [17]. Other mutations, albeit less frequently observed, occur in the DNMT3, TET2, and IDH1/2 genes [18]. *EVI1*r AML often presents with monolobated megakaryocytes, multilineage dysplasia, and normal/elevated blood platelet counts [19].

**−5 or del(5q); −7; −17/abn(17p):** These abnormalities are commonly observed in patients with AML, previously treated with chemotherapy, including alkylating agents, platinum-based agents, or antimetabolites. 5q deletion is typically large, involving ∼70 Mb of the 5q14-q33 chromosome. This region includes haploinsufficient genes like *RPS14* (ribosomal protein S14) and *APC* (adenomatous polyposis coli), microRNA genes (mir-145 and mir-146A), which are implicated in megakaryocytic dysplasia, as well as genes controlling hematopoietic stem cell expansion, such as *EGR1* and *CSNK1A1* [20]. Monosomy 7, the most common autosomal monosomy in AML, and frequently seen in therapy-related AML [20], can be also found in congenital diseases predisposing to myeloid neoplasms, such as those bearing germline GATA2 mutations, or affected by neurofibromatosis, and severe congenital neutropenia [21]. The tumor suppressor genes located in chromosome 7 are believed to act in a haploinsufficient manner, and include *SAMD9/SAMD9L* endosomal proteins, *EZH2* histone modifying enzyme, and MLL3, which is associated with *Ras* pathway mutations and *TP53* inactivation [21]. 17p deletion or monosomy commonly involves the tumor suppressor gene p53 on band 17p13.1.

**Complex karyotype (CK)**: CK is defined by the presence of ≥3 chromosomal abnormalities in the absence of specific recurring translocations or inversions included in the WHO classification [22], such as t(8;21), inv(16) or t(16;16), t(9;11), t(v;11)(v;q23.3), t(6;9), and inv(3) or t(3,3) [23]. This subtype accounts for 10–12% of adult AML cases, with the most common chromosomal losses being the 5q (80% of cases), 7q, and 17p chromosomes [24]. More recently, CK AML has been proposed to be further subclassified into typical CK-defined by the presence of 5q and 7q abnormalities and/or 17p loss- and atypical CK, which lacks these specific chromosomal abnormalities. Typical CK AML, often associated with *TP53* mutations (in 80% of cases), has very poor prognosis [24]. In contrast [25], patients with atypical CK AML, who are generally younger, frequently have mutations in *PHF6*, FLT3-TKD, MED12, and NPM1, and tend to achieve a longer overall survival compared to those with typical CK AML [24]. 

**Monosomal karyotype (MK)**: MK is defined by the presence of ≥2 distinct autosomal monosomies or a single autosomal monosomy accompanied by structural abnormalities (deletions of -X or -Y are not considered monosomies) [26]. MK AML occurs more frequently in therapy-related cases compared to de novo AML and is closely associated with alterations in the *TP53* gene, leading to significant chromosomal instability [27]. The most common chromosomal alterations include monosomy 7 (∼35%), monosomy 5 (∼22%), and −17 (∼11%) [27]. 

**Mutated RUNX1**: *RUNX1* mutations typically affect the Rnt Homology Domain (RHD) or the Transactivation Domain (TAD) of the gene (located at 21q22), which encodes the alpha subunit of the Core Binding Factor (CBF). Given the association of RUNX1 mutations with autosomal-dominant thrombocytopenia, it is advisable to screen for germline mutations among family members to rule out this hereditary condition. *RUNX1*-mutated AML is predominantly observed in patients who are older males. It may be preceded by Fanconi anemia or congenital neutropenia. A prior history of myelodysplastic syndrome or prior exposure to radiation can be present. There is a frequent association with *MLL-*PTD or *ASXL1* mutations [28,29], indicating a complex genetic landscape that influences disease progression and treatment response.

**Mutated EZH2**: Enhancer of Zeste Homolog 2 (EZH2) is a key component of the polycomb group (PcG) proteins, which are crucial for gene silencing via histone modifications [30]. EZH2 composes the regulatory hub of PRC2, which functions as a histone H3 lysine 27 methyltransferase [30]. Unlike its role in clonal hematopoiesis (CH), where EZH2 mutations are not typically implicated, these mutations are more commonly associated with the development of overt leukemia [31]. EZH2 mutations could be initiating an event or occur later on during leukemogenesis to drive clonal expansions [31]. The prevalence of EZH2 mutations in de novo AML ranges from 1 to 4% of patients [32,33,34]. The *EZH2* gene is located at 7q36.1, a genomic region that is often deleted in AML (−7 or del7q), and associated with an adverse prognosis. In AML, *EZH2* frequently undergoes nonsense and frameshift mutations, leading to its inactivation. Notably, mutations in the serine and arginine rich splicing factor 2 (*SRSF2*), which is a high-risk genomic entity [1], could affect EZH2 expression by modifying the sequence-specific RNA binding activity of EZH2. This, in turn, alters the recognition of splicing enhancer motifs, leading to aberrant EZH2 splicing and nonsense-mediated decay and decreasing the expression of EZH2, thereby influencing H3K27me3 levels. Furthermore, mutations in the ASXL1 gene, another polycomb-related protein mutated in HR-AML [33], also decrease H3K27me3 levels by impairing PRC2 recruitment. These mechanisms contributes to the activation of *HOXA9*-driven leukemogenesis [35]. In myeloid neoplasms, *EZH2* mutations tend to be mutually exclusive with *SRSF2* and *U2AF1* mutations [36], while it is more frequently co-mutated with *ASXL1* and *TET2* [36,37]. 

**Mutated ASXL1**: *Additional sex combs-like 1* (*ASXL1*) is a critical epigenetic modifier, whose mutations are commonly identified in CH [38,39,40]. In murine models, ASXL1 knockdown leads to a myelodysplastic-like phenotype, primarily due to the loss of interaction with PRC2 [35,41,42,43]. In myeloid neoplasms, the majority of ASXL1 mutations consist of frameshift or nonsense mutations at exon 12. These mutations are mutually exclusive with *DNMT3A*, *FLT3*-ITD, and *NPM1* mutations, while ASXL1 mutations frequently co-occur with mutations in DNA methylation genes (such as *TET2* and *IDH1-2*), spliceosomes (*U2AF1*, *SRSF2*), transcription factors (*CEBPA*, *RUNX1*, *GATA2*), and signal transducers (*NRAS*, *JAK2*, *STAG2*) [44]. In AML, the frequency of ASXL1 mutations is about 5–10% [33,45], with a higher prevalence in patients who are older and those with secondary AML. *RUNX1* is the most frequent co-mutated gene and cooperates with mutant ASXL1 to support myeloid leukemogenesis in vivo [46]. 

**Mutated BCOR**: The *BCL6 corepressor* (*BCOR*) is a tumor suppressor gene, which is dysfunctional in lymphoid and myeloid tumors [47]. BCOR is a critical component of the noncanonical PRC1.1, which is recruited to specific chromatin regions in a context-specific manner [47]. Mutations of BCOR are detected in about 5% of adult de novo AML and 4% of AML with myelodysplasia-related changes [33,48]. The frequency of *BCOR* mutations is even higher in secondary AML [49]. Most commonly, patients with BCOR-mutated AML carry a normal karyotype (NK). In AML with NK, about 45% of *BCOR*-mutated AML have co-mutations with *DNMT3A* and/or *RUNX1*, while being mutually exclusive with *NPM1* and *FLT3* mutations [50,51]. Patients with *BCOR* mutations usually have activated RAS signaling, due to the high rate of *NRAS* and *KRAS* mutations [47]. In vivo, *BCOR* leads to overt acute leukemia in the presence of co-mutations, such as *DNMT3A* [51] or *RAS* mutations [52].

**Spliceosome mutations (SRSF2, SF3B1, U2AF1, ZRSR2):** The most commonly mutated genes in this category are splicing factor 3B subunit 1 (*SF3B1*), serine and arginine rich splicing factor 2 (*SRSF2*), U2 small nuclear RNA auxiliary factor 1 (*U2AF1*), and zinc finger, CCCH type, RNA-binding motif, and serine and arginine rich 2 (*ZRSR2*) [33], which are implicated in the early assembly of the spliceosome machinery [53]. Mutations in splicing factors (SFmut) are predominantly early events in leukemogenesis [54]. Mutations in splicing factors account for about 18% of adult AML [33], are more frequent in older age, and commonly are associated with multilineage dysplasia [55]. While mutations of *SF3B1*, *SRSF2,* and *U2AF1* are gain-of-function, determining a change of amino acid residues [56], mutations of *ZRSR2* are inactivating nonsense or frameshift [56]. Mutations in SF are always heterozygous and mutually exclusive with each other [56].

However, the pattern of co-mutations between *STAG2*, *RUNX1*, *SRSF2*, and *ASXL1* (SRSA genes) [57] or between *SRSF2* and *IDH2* [56] have been described in human AML. In mice, *SF3B1*, *U2AF1*, and *SRSF2* mutations cause aberrant hematopoiesis and the acquisition of myelodysplastic-like phenotypes [58,59,60,61]. The mechanisms of splicing factors’ dysregulation in myeloid leukemogenesis have been extensively reviewed [62]. Briefly, several studies have analyzed the impact of mutations of specific splicing genes and implications for leukemogenesis: (i) mutations in SRSF2 and U2AF1 yield alternative exon usage; (ii) the ZRSR2 mutant induces the retention of minor introns (U12-type) [63]; (iii) the SF3B1 mutant instigates the usage of alternative branch points to cause an alternative 3′ splice site [64,65]. SF mutations induce mis-splicing of hematopoietic regulators, such as *EZH2* in *SRSF2*-mutated MDS [58].

**Mutated STAG2 (cohesin complex)**: Mutations in the cohesin subunit SA-2 (STAG2) define AML with myelodysplasia-related gene mutations, irrespective of prior MDS [1], and are considered a marker of poor prognosis. STAG2, together with double-strand-break repair rad21 homologue (RAD21), and the structural maintenance of chromosomes (SMC1A and SMC3) form the core of the cohesion complex, which surrounds sister chromatids during replication, and support the transition from metaphase to anaphase [66]. The roles of cohesin mutations in leukemogenesis are multiple, as they can induce aneuploidy through mis-segregation of sister chromatids or remodel 3D chromosome topology and chromatin interactions [66]. In vivo, mutated cohesion subunits induce the acquisition of a pre-leukemic phenotype, with altered erythroid and myeloid lineages’ differentiation. Mutations in the cohesion genes range between 6 and 13% in AML [67,68], are mutually exclusive, and can be accompanied by NK or CK. Most *STAG2* mutations are nonsense or frameshift, leading to protein truncation or loss-of-function [63]. *STAG2* mutations are often, if not always, associated with *RUNX1*, *SRSF2*, and *ASXL1* mutations [63]. Although *STAG2* mutations are classified within the adverse-risk category, their prognostic significance appears to be linked to the presence of other co-mutations. When multivariate analyses are adjusted for mutation in *BCOR*, *ASXL1*, and *RUNX1*—which are more commonly found in *STAG2*-mutated AML compared to other subsets-STAG2 mutations—they lose their independent prognostic impact. Intriguingly, mutated STAG2 significantly increases the sensitivity of AML cells to poly ADP-ribose polymerase (PARP), such as talazoparib [69,70]. This suggests that the presence of STAG2 mutations could potentially be exploited to tailor more effective therapeutic strategies in this setting.

**Mutated TP53**: The majority of *TP53* mutations are missense, with hotspots in arginine residues, though other mutational events have been reported, including insertions, deletions, and frameshift mutations. More frequently, the mutation occurs in the DNA binding domain, with the loss of function of the p53 tumor suppressor, despite some mutations being able to lead to gain-of-function through the binding of mutant p53 to other tumor suppressors such as p63 and p73 [71]. The frequency of *TP53* mutations in de novo AML ranges from 5 to 10%, increasing to approximately 30% in cases of therapy-related AML and AML with complex cytogenetics. *TP53* mutations are particularly prevalent in AML cases that exhibit CK, chromothripsis, or monosomal karyotype [72]. Interestingly, TP53 mutations are less commonly found with mutations in *DNMT3A*, *TET2*, and *IDH1-2* [72]. Moreover, the variant allele frequency of *TP53* appears to be directly correlated with the level of cytogenetic complexity and inversely correlated with overall survival in patients with AML [73].

## 3. Immune Landscapes of AML with Adverse Genetics

HR-AML is distinguished by elevated inflammation (as indicated by a high iScore), greater clonal diversity, and a higher immunogenic potential [74]. AML harboring *TP53*, *RUNX1*, *ASXL1*, and *RAS* mutations, found in the adverse-risk category, exhibits a higher immune effector dysfunction (IED172) score and an IFNγ signature, the latter being associated with a positive response to azacytidine (AZA) + pembrolizumab [75]. AML with mutated *TP53* is characterized by enrichment for gene programs related to T-cell lineage commitment, positive T-cell selection, and T-cell homeostasis, indicating a T-cell-rich environment, as well as for an IFNγ-dominant tumor microenvironment (TME) [76]. *TP53*-mutated AML is also enriched for the tumor inflammation signature (TIS), as well as characterized by the upregulation of immune checkpoints such as *PD-L1*, *TIGIT*, and *LAG3* and markers of immune senescence [77]. Interestingly, PD-L1 upregulation is mostly restricted to HSCs in TP53-mutated AML, while T-cell immunity features low levels of PD-1 on CD8+ cytotoxic T-cells and an expansion of ICOS^hi^/PD1^−^ Tregs [78]. Furthermore, AML with a higher number of mutations or HR-AML is more infiltrated by immune cells and has higher expression of *PD-L1*, *FoxP3*, *GzmB*, *PTEN*, and *BCL2* genes, as well as of gene networks linked to immune-exhaustion [76]. Importantly, patients with immune-infiltrated AML and adverse ELN characteristics derive significant benefit from allo-HCT [76]. The cytolytic score (geometric mean of *GZMA*, *GZMH*, *GZMM*, *PRF1*, and *GNLY*) correlates with *TP53* mutations and the deletion of chromosome 5, in AML [79]. An analysis of the Hemap AML and BeatAML datasets has shown that cases with a high cytolytic score are characterized by an MDS-like phenotype with complex cytogenetics and a history of MDS [79]. The cytolytic score correlates with the diagnosis of AML with myelodysplasia-related changes, suggesting a link between an MDS-like/sAML subtype and an increased cytolytic infiltration. The MDS-like subtype has been associated with *RUNX1*, *TP53*, *U2AF1*, and *SRSF2* mutations. Leukemic blasts from MDS-like AML more frequently are classified as HSC or progenitor-like cells, such as multipotent progenitors, megakaryocyte-erythroid progenitors, or granulocyte–monocyte progenitors. Furthermore, AML with a higher cytolytic score has a higher infiltration of NK and CD8+ T-cells, the latter biased toward a cytotoxic and effector-memory phenotype [79]. These results suggest that the leukemia cell state of differentiation may influence the composition of the bone marrow microenvironment, as well as the interactions between immune cells. MDS-like AML blasts have higher expression of HLA-II, LGALS9, and TGFB1, while T- and NK cells display elevated levels of their cognate receptors *LAG3*, *HAVCR2*, and *TGFBR3* and secrete more IFNγ, compared to non-MDS-like AML [79]. MDS-like AML more frequently expresses *CD274* and *ARG1* inhibitory genes and their corresponding receptors [79].

## 4. Immunotherapy in AML with Adverse Genetics

Immunotherapy strategies for treating AML have been extensively reviewed elsewhere [80,81,82]**,** and providing a detailed overview of each type of immunotherapy approach falls outside the scope of this review article. Immune-based treatments against AML, which are currently being investigated in clinical trials both as monotherapy or in combination with anti-leukemic drugs such as hypomethylating agents, mostly include monoclonal antibodies and immune checkpoint blockade (ICB). These latter approaches will be reviewed in the next sections. In addition, other therapeutic platforms such as T-cell engagers and cancer vaccines are reporting encouraging anti-leukemic activities in early trials [83]. In this regard, T-cell engagers, including Bispecific T-cell engagers (BiTE), which have affinities for both a tumor antigen and an immune cell receptor, are currently being tested as monotherapy strategies in relapsed/refractory (R/R) AML. The CD123 × CD3 DART flotetuzumab has shown a composite complete response (including complete response/CR, CRi defined as incomplete blood recovery, and morphologic leukemia-free state/MLFS) of 20%, with an objective response rate/ORR of 24% in patients with R/R AML [84]. The anti-leukemia activity of a second-generation CD123 × CD3 DART (MGD024) is currently being investigated in an ongoing clinical trial (NCT05362773). The phase 1 dose escalation trial evaluating CD33 × CD3 BiTE (AMG330) has registered a cCR rate of 19% in R/R AML patients [85]**,** indicating the need to investigate potential combinatorial approaches with other drugs to achieve a more effective and durable response in this setting. To this end, in preclinical experiments, AMG330 has been shown to favor the upregulation of immune checkpoints, such as PD-L1, on target and effector cells. The blockade of the PD-1–PD-L1 interaction significantly enhanced AMG330-induced leukemia cell lysis, T-cell proliferation, and IFNγ secretion [86]. Another emerging immunotherapeutic strategy against AML is represented by cancer vaccination, which can be mainly categorized into peptide vaccines and dendritic cell (DC)-based vaccines [87]. The main goal of AML vaccines is to elicit an effective cellular and/or humoral immune response. To this end, leukemia-associated antigens (LAAs) should be highly expressed, immunogenic, and (mainly) restricted to leukemic cells [87]. LAAs more extensively investigated in phase I or II trials are Wilms tumor (WT1), mucin 1 protein (MUC1), proteinase 3 (PR3), and the receptor for hyaluronic acid-mediated motility (RHAMM), which is found overexpressed in >80% of AML patients [88]; based on the results from early trials, the setting that more likely could benefit more from interventions with peptide vaccines is represented by patients in CR or with minimal residual disease. DC-based vaccines are based on exploiting the biological functions of DCs being proficient at stimulating both innate and adaptive immune responses. Clinical trials with DC vaccination have been mainly directed against AML relapse in the presence of minimal residual disease (MRD) persistence. Also in the case of DC vaccines, WT1, along with other LAAs, represents one of the most targeted antigens [89]. Encouraging results have been obtained with next-generation DC-based immunotherapy loaded with WT1 or PRAME and cultured with TLR agonists [90]. An important aspect to be taken into account when engineering DC-based vaccines is represented by the cell-of-origin designated for vaccine production. Indeed, DCs could originate from leukemic cells (AML-DCs) and monocytes (mo-DCs). AML-DCs have several advantages, as they can directly present LAAs along with MHC molecules on the cell membrane, while mo-DCs need to be loaded with LAAs [87]. Immunotherapy with peptide or DC vaccines is a rapidly growing field of investigation, and will mostly benefit from large-scale proteomic studies characterizing the AML surfaceome, which are aimed at the identification of highly expressed AML-restricted antigens. In addition, other important aspects to be further addressed in this field of immune-based therapeutics are (i) the definition of biological rationales for combinatorial approaches adding vaccines to other drugs, such as ICB or monoclonal antibodies, and (ii) the timing of intervention, as AML vaccines could be highly efficacious in eradicating MRD persistence after allo-HCT, or in preventing AML onset by targeting high-risk preleukemic states, such as CHIP or myelodysplastic syndromes.

## 5. Rationales for Combining Targeted Therapy with Immunotherapy in AML with Adverse Genetics

While immunotherapy, particularly ICB, offers a promising strategy to stimulate the immune system’s natural ability to fight cancer, its effectiveness as a monotherapy in AML is limited, benefiting only a subset of patients [83]. This limitation emphasizes the need to search for rational synergistic approaches adding immunotherapy to other treatment modalities that are able to modulate immune escape mechanisms active in the leukemic microenvironment, as well as for biomarkers of response to ICB. It is indeed possible to achieve a more robust and durable antitumor response by integrating targeted therapies that directly inhibit the oncogenic drivers, or that re-educate non-malignant components of the leukemia microenvironment (i.e., BM stroma) with immunotherapies that enhance the immune system’s capacity to detect and destroy cancer cells. 

### 5.1. Exploiting BCL2 Inhibition for Innate and Adaptive Immune Reactivation

The combination of AZA with venetoclax (VEN) has achieved complete response (CR)/CR with incomplete count recovery (CRi) rates of 70% in patients with adverse-risk cytogenetic AML without *TP53* mutations, as well as durable remission (18.4 months) and improved OS (23.4 months) [91]. Unfortunately, these results cannot be extended to AML cases with *TP53* mutation, where the response rate and overall prognosis remain poor and comparable to the historical results with HMA. Other high-risk genotypes that are sensitive to BCL2 inhibition are those harboring *ASXL1* [92] and *RUNX1* mutations [93]. Indeed, hematopoietic stem/progenitor cells from patients with *ASXL1*-mutated AML have a higher expression of *BCL2* [94], and relapsed–refractory *ASXL1*-mutated AML treated with HMA and VEN had improved CR/CRi rates in a retrospective study [92]. Recent evidence has proven that, beyond direct anticancer effects, BCL2 inhibition is linked to broader immunomodulatory functions: (i) BCL2 inhibition activates dendritic cells to enhance antitumor immunity and sensitize tumors to anti-PD(L)1 immunotherapy (Figure 1) [95]; (ii) the Bcl2 inhibitor, VEN, has been shown to increase the effector activity of antileukemic T-cells without inducing T-cell apoptosis (Figure 1), through reactive oxygen species release, against AML in vitro and in vivo [96]; (iii) VEN can augment the antitumor efficacy of ICB, as it increase the frequency of PD1+ effector-memory T-cells in mouse tumor models [97].

### 5.2. Targeting TP53-Dependent or -Independent Mechanisms of Apoptosis with APR-246/Eprenetapopt

APR-246/eprenetapopt is a small molecule that targets TP53-mutated cancers [98,99], which has shown promising results against *TP53*-mutated MDS and AML [100,101,102]. APR-246 reactivates mutant p53 transcription, by facilitating its binding to DNA sequences, eventually inducing apoptosis [99]. APR-246 can also cause tumor cell death in p53-independent mechanisms, as for instance by impairing the balance between glutathione (GSH) and reactive oxygen species [103,104]. More recently, using AML cell lines and leukemia xenografts, it has been shown that APR-246 depletes intracellular GSH and induces lipid peroxide production, eventually leading to the induction of ferroptosis [105]. Ferroptosis is a programmed cell death induced by iron-dependent lipid peroxidation [106]. Importantly, chemotherapy-resistant tumor cells can be instead greatly sensitive to ferroptosis [107], as might be the case of HR-AML. Ferroptosis may exert a double-edge function in the tumor microenvironment, by activating or suppressing immunity. 

Thus, searching for cancer-specific correlations between ferroptosis induction and the microenvironmental dependence on immunostimulatory or immunoinhibitory checkpoints is key to designing rational combinatorial approaches. In this regard, the following has been reported: (i) ferroptosis may dampen immune tolerance by inducing the death of glutathione peroxidase (GPX4)-deficient Tregs trough CD28 costimulation [108]; GPX4 is the key regulator of ferroptosis, since it interrupts the lipid peroxidation chain reaction [109]; (ii) CTLA4 expression is higher in tumors with higher ferroptotic scores (Figure 1) [107,110]; (iii) ferroptosis can inhibit tumor immune tolerance by recruiting the ATP-P2X7-CD86 axis [107]; (iv) immunotherapy-activated CD8+ T-cells enhance ferroptosis-specific lipid peroxidation in tumor cells, contributing to cancer immunotherapy efficacy [111]; (v) early ferroptotic cells undergo immunogenic cell death, associated with the release of damage-associated molecular patterns (DAMPs) and an enhanced maturation of dendritic cell [112]. Even though experimental insights are currently lacking in AML models, the combination of ferroptosis-inducing agents such as APR-246/eprenetapopt, which is promising in treating *TP53-*mutated AML [101,102], may benefit from the combination with ICB, as anti-CTLA4 or anti-PD(L)1 (Figure 1).

### 5.3. Targeting CD47 Phagocytic Immune Checkpoint in Adverse-Risk AML

CD47 plays a crucial role in the evasion of phagocytosis by AML cells [113]. Its overexpression is associated with a poorer prognosis [114]. Preclinical evidence has found that targeting CD47 with the humanized anti-CD47 antibody magrolimab might represent an effective strategy to treat AML [115]. Magrolimab is a first-in-class investigational monoclonal antibody against CD47 and macrophage checkpoint inhibitor, which interferes with the recognition of CD47 by the SIRPα receptor on macrophages, thus blocking the “don’t eat me” signal used by cancer cells to evade phagocytosis (Figure 1). Several clinical trials are currently ongoing to search for AML patients who could benefit more from anti-CD47/SIRPa immunotherapy. Recent findings also suggest that CD47 expression in AML is genotype-dependent, with higher antigenic density observed in cases with CBFB/MYH11 rearrangements or NPM1 mutations. Conversely, AML with adverse risk genetics, such as MLL-rearranged AML, shows less consistent CD47 expression, with some cases nearly negative for CD47 on leukemic blasts. These findings underscore the potential of personalized approaches that might combine CD47-targeting therapies with agents that can increase CD47 expression or enhance “eat me” signals, such as HMA [116]. 

### 5.4. Targeting Poly(ADP-Ribose) Polymerase in STAG2-Mutated AML

AML with mutated *STAG2* appears more sensitive to PARP inhibitors, which inhibit the DNA damage response (DDR), thereby increasing the neoantigen load and mutational burden. PARP inhibitors can generate tumor-derived double-stranded DNA in the cytoplasm, which is sensed by the cytosolic DNA sensor cyclic GMP-AMP synthase, thus activating the stimulator of interferon (IFN) genes (STING) signaling pathway [117]. STING activation induces the upregulation of type I IFNs, which promote systemic immune response. PARP inhibitors can reprogram the tumor immune microenvironment by sustaining a Th1 immune response and can upregulate PD-L1 expression through GSK3β inactivation [117] (Figure 1). Of note, cohesin (STAG2)-mutated cancers have been reported to display strong activation of IFN and NF-kB expression signatures, along with PD-L1 upregulation [118], thus providing another rationale for adding anti-PD(L)1 immunotherapy in *STAG2*-mutated AML. In advanced solid tumors, the anti-PD-L1 avelumab has been recently combined with talazoparib with evidence of better responses in *BRCA-*altered tumors [119]. Given that cohesin directly regulates the DNA damage checkpoint activation and repair pathways and that tumors deficient in DNA damage response achieve durable benefit from ICB [117], *STAG2*-mutated AML might represent a promising subset for immunotherapy with ICB.

### 5.5. Splice-Site-Creating Mutations and Sensitivity to Immune Checkpoint Inhibition

Tumors harboring splice-site-creating mutations (SCMs) generate more neoepitopes than non-synonymous mutations and possess a higher expression of PD-L1 (compared to tumors without SCMs) [120]. This characteristic is of importance considering that an augmented generation of neoantigens can lead to enhanced efficacy of ICB in tumors with low immunogenicity [121], such as AML. Further reinforcing this evidence, recent bioinformatic analyses have identified that a specific set of splicing mutations correlates with poor prognosis, increased infiltration by myeloid cells with suppressive phenotypes, and elevated expression of immune checkpoints in the leukemic microenvironment. These preliminary observations suggest that AML harboring SCMs could be particularly susceptible to ICB [122].

## 6. Current Treatment Strategies for AML with Adverse Genetics

Based on the recent ELN guidelines [1], the eligibility for standard intensive chemotherapy depends primarily on the fitness of the patient, based on age and comorbidities [1]. Patients who are fit, with HR genetics and no targetable lesions, are mainly treated with a standard regimen based on anthracyclines and cytosine arabinoside. These patients, especially with TP53 mutations [123], cannot benefit from the addition of the CD33 inhibitor gemtuzumab [124], nor from the use of encapsulated anthracycline-AraC molecules (CPX 351). For patients who respond to induction chemotherapy, allo-HCT remains the only potentially curative treatment because of the immunological effect of the graft versus leukemia [125], and subsequent post-HCT immunomodulatory treatments such as donor lymphocyte infusions or specific drugs could be beneficial in this high-risk population. However, even if recent improvements in allo-HCT platforms appear encouraging [126], outcomes remain unsatisfactory especially in TP53-mutated AML, with an OS of less than 30% at 2 years [127].

### Venetoclax Plus Azacytidine

For patients unfit for intensive chemotherapy, VEN + AZA are now considered the standard front line treatment based on the results of the Viale A trial [128]. Of note, for patients with adverse-risk genetic mutations, given the poor prognosis associated with intensive chemotherapy, there has been interest in less intensive targeted therapeutic approaches. 

Recently, Pollyea et al. [91] analyzed the outcomes of 127 AML patients with HR genetics treated with AZA + EN in front-line treatment compared to 56 patients treated with AZA alone. The combination of AZA + VEN in patients with adverse genetics allowed achieving a complete remission rate in 70% of patients versus 30% for AZA alone, with a median OS of 23 months versus 11.3 months, respectively. Importantly, the outcomes of patients treated with AZA + VEN were comparable to patients who were similarly treated with intermediate-risk cytogenetics. However, for patients with Tp53 mutation, even if CR was achieved in 41% with AZA + VEN versus 17% with AZA alone, no benefit was observed in OS (5.2 months versus 4.9 months). 

The use of AZA + VEN is of interest also in the specific context of several adverse genetic mutations. In particular, a retrospective study, conducted by Aldoss at al. [92], reported outcomes for 90 patients with relapsed–refractory AML treated with AZA + VEN. The presence of ASXL1 mutation or TET2 was associated with better response. Furthermore, the association of ASXL1 with a better response to AZA-VEN was recently confirmed in the setting of MDS [129].

However, a more recent study conducted by Cherry et al. [93], which retrospectively compared patients with newly diagnosed AML who received AZA + VEN (*n* = 143) versus intensive chemotherapy (*n* = 149), did not confirm the better results for ASX L1 mutations, but showed that RUNX 1 mutations could benefit from the combination of AZA-VEN as first-line treatment. 

The mutational testing pre-treatment will be more and more important in treatment planning, but more data are needed to choose the best treatment in HR AML. Novel treatment combinations are needed to improve remission rates, and also, recent guidelines [1,130] reflect the need for novel treatment approaches, including the combination of target and immunomodulatory agents. 

## 7. Promising Targeted Approaches for the Treatment of AML with Adverse Genetics

**Menin inhibitors** are compounds that disrupt the interaction between the scaffolding protein menin and the methyltransferase KMT2A. Among these inhibitors, revumenib (SNDX-5613) stands out as one of the most prominent, while others like JNJ-75276617 and KO539 show considerable promise in ongoing development efforts. Revumenib is recognized for its potency and selectivity as a small molecule that effectively disrupts the interaction between menin—a crucial scaffold protein—and histone-lysine N-methyltransferase 2A, encoded by the KMT2A gene. Together, these proteins regulate gene expression through epigenetic mechanisms. Certain genetic alterations, such as KMT2A rearrangement and NPM1 mutation, can disrupt the proper regulation pf epigenetic programs, leading to an aberrant proliferation of leukemia cells. Menin inhibitors like revumenib bind to menin, effectively halting this aberrant process and restoring normal blood cell production. More recent milestones include revumenib’s Orphan Drug Designation from both the FDA and the European Commission for treating AML. Additionally, it has received Fast Track designation from the FDA for treating relapsed–refractory acute leukemias in both adult and pediatric patients who harbor KMT2A rearrangement or NPM1 mutation. These designations underscore the urgent need for innovative treatments in these specific patient populations and emphasize revumenib’s potential as a promising therapeutic option in the management of AML. 

Another interesting targeted approach includes the use of **anti-CD123-directed therapies**. CD123 is a subunit of the interleukin 3 (IL3) receptor expressed on the surface of blasts in most AML cases, particularly in poor-risk genetic subgroups. CD123 expression is associated with high cell count at diagnosis and poor prognosis. Tagraxofusp (SL-401) is a recombinant protein targeting CD123 and is currently approved as monotherapy for the treatment of blastic plasmacytoid dendritic cell neoplasm (BPDCN). Additionally, Pivekimab Sunirine (PVEK, IMGN632) is an antibody-drug conjugate (ADC) consisting of a high-affinity CD123 antibody, a cleavable linker, and an indolinobenzodiazepine pseudodimer (IGN) payload. Flotetuzumab (MGD006) is a bispecific antibody engineered to bind CD3 and CD123 on AML cells. Both PVEK and flotetuzumab are being investigated as monotherapies and as combination therapies for AML. These agents hold promise in targeting CD123-expressing AML cells and may offer new treatment options for patients with this challenging disease. Further, given that current immunotherapeutic strategies, especially as monotherapy, appear to only slightly ameliorate the outcomes of patients with HR-AML, novel treatment modalities should be urgently investigated in preclinical research and early clinical trials. Among these, emerging approaches that are promising for treating aggressive tumors, including AML, are (i) engineered nano-theranostic materials (e.g., metal-based agents, or bio-inspired nanoparticles), which can achieve a durable release of the therapeutics with low off-target toxicities [131,132], while also eliciting temporally and spatially defined immune responses against the leukemic clone [133], and (ii) CRISPR-based therapies targeting fusion oncoproteins [134] or molecules selectively expressed by leukemic stem cells [135]. 

## 8. Novel Investigational Strategies Combining Immunotherapy and Target Therapy in AML with Adverse Genetics

The clinical trials described in this section are summarized in Table 2. 

### 8.1. APR-246-Based Combinations 

The first clinical trial that investigated the combination of APR-246 and AZA was a U.S. phase II trial [102,136] (NCT03072043) in which there were 55 patients with TP53 mutation (40 MDS and 11 AML) with a median age of 66 years enrolled. The overall response rate (ORR) was 71% with a CR rate of 44%, and 38% achieved MRD negativity assessed by NGS. The median duration of CR was 7.3 months, with a median follow-up of 10.5 months. The median OS was 10.8 months. A French phase II trial [102] (NCT03588078) enrolled 52 patients (34 MDS and 18 AML) with a median age of 74 years. The ORR rate was 52% with a CR rate of 37% with 30% of patients with MRD negativity. The median duration of CR was 11.7 months, with a median follow-up of 9.7 months. The median OS was 12.1 months. No additional hematological toxicity was reported compared to AZA alone. However, neurological effects including ataxia, acute confusion, facial dizziness, and paresthesias were reported in 40% of patients. Based on these results, a phase III randomized clinical trial was conducted to compare AZA alone + AZA + APR-246 in MDS (NCT03745716). The results failed to demonstrate the superiority of the combination compared to AZA alone. However, more recently, a phase I trial (NCT04214860) has shown that the addition of APR-246 to VEN and AZA appears encouraging in treating *TP53*-mutated AML with a well-tolerated toxicity profile and promising efficacy by achieving an overall response of 64% (25/49) and a CR of 38% (15/39) [137]. Furthermore, APR-246 has been investigated in the post-HCT setting (NCT03931291) [101]. There were 33 patients (14 AML and 19 MDS) with m*TP53* who received post-HCT maintenance treatment with up to 12 cycles of eprenetapopt 3.7 g once daily intravenously on days 1–4 and AZA 36 mg/m^2^ once daily intravenously/subcutaneously on days 1–5 in 28-day cycles. The median number of eprenetapopt cycles was 7 (range, 1–12). With a median follow-up of 14.5 months, the median relapse-free survival (RFS) was 12.5 months and the 1-year RFS probability was 59.9%. With a median follow-up of 17.0 months, the OS was 20.6 months and the 1-year OS probability was 78.8%. Acute and chronic (all grade) graft-versus-host disease and adverse events were reported in 12% (*n* = 4) and 33% (*n* = 11) of patients, respectively.

### 8.2. Innate and Adaptive Immune Checkpoint Inhibition in AML with Adverse Genetics

**Magrolimab (anti-CD47)**: Daver et al. recently published the results of a phase Ib trial (NCT03248479) investigating the safety and efficacy of magrolimab in association with AZA in previously untreated AML ineligible for chemotherapy [138]. Eighty-seven patients were enrolled: 82.8% had *TP53* mutations; fifty-seven (79.2%) of patients with the *TP53* mutant had adverse-risk cytogenetics. Patients received a median of four cycles of treatment. Each cycle consisted of infusion of magrolimab as an initial dose (1 mg/kg, days 1 and 4), followed by 15 mg/kg once on day 8 and 30 mg/kg once weekly or every two weeks as maintenance. Azacitidine 75 mg/m^2^ was administered intravenously/subcutaneously once daily on days 1–7 of each 28-day cycle. The most common treatment-emergent adverse events included constipation, nausea and diarrhea, and anemia. There were 32.2% of the patients who achieved CR, including 31.9% patients with *TP53* mutations. The median OS in patients with the *TP53* mutant and wild-type were 9.8 months and 18.9 months, respectively. Based on these results, new phase III randomized clinical trials are recruiting front-line patients. ENHANCE-2 (NCT04778397) is investigating the role of magrolimab plus AZA versus Physician’s Choice of VEN-AZA or intensive chemotherapy in patients with TP53 AML in previously untreated AML and ENHANCE-3 (NCT05079230) the role of magrolimab versus placebo in combination with venetoclax and Azacitidine in previously untreated patients with acute myeloid leukemia ineligible for intensive chemotherapy.

**Sabatolimab (mb5-453):** T-cell immunoglobulin domain and mucin domain-3 (TIM-3) is a T-cell immune checkpoint that regulates adaptive and innate immunity and is aberrantly expressed on the surface of leukemic cells, and higher levels of expression are associated with poor prognosis [139]. Sabatolimab, a novel anti-TIM3 monoclonal antibody, exerts its antileukemic activity by a direct targeting of TIM-3 on the blast surface, promotes antibody-dependent phagocytosis, and promotes the block of TIM-3–GALAECTIN-9 interaction, preventing leukemia stem cell renewal [140]. Sabatolimab has been investigated in association with HMA in patients with HR-MDS and AML unfit for intensive chemotherapy. The patients with AML were 48. The ORR was 40%, and of these, 30% achieved CR. The median duration of response was 12.6 months with a PFS of 27.9%. For patients with at least one genetical adverse-risk mutation, the ORR was 53.8% with a median duration of response of 12.6 months [141]. Based on these results, the STIMULUS clinical trial program was started in which randomized phase II and phase III clinical trials are investigating multiple combinations with sabatolimab based in AML, high-risk MDS, and chronic myelomonocytic leukemia. STIMULUS-AML1 (NCT04150029) is an ongoing phase II, single-arm study of sabatolimab + AZA + VEN in adult patients with AML ineligible for intensive chemotherapy [142].

**Nivolumab:** Nivolumab is an antibody that binds to PD-1 and blocks signaling mediated by PD-1–PD-L1 interactions. Also, nivolumab blocks signaling mediated by PD-1–PD-L2 interactions. Nivolumab is used to treat various cancers such as melanoma, Hodgkin’s lymphoma, and non-small cell lung cancer (NSCLC). A phase II trial (NCT02397720) assessed the efficacy and safety of nivolumab in combination with AZA in 70 patients with relapsed–refractory AML. The ORR was 33%, of which, 22% achieved CR with a median OS of 6.3 months. Responses were higher in patients not pretreated with HMA (ORR: 52%) [143], and *ASXL1* mutations were associated with improved ORR and OS. Upregulation of CTLA-4 expression on T-cells was observed in patients that did not achieve remission, suggesting that CTLA-4 overexpression could be a potential mechanism of the resistance of PD1 blockade [143]. So, a subsequent cohort was added (36 patients) and treated with Ipilimumab (antiCTLA-4) + AZA+ nivolumab with the aim to enhance T-cell response. The ORR was 46%, of which 36% achieved CR. The median OS was 10.5 months and comparably better with AZA + nivolumab. Two new ongoing clinical trials are further investigating the role of these combinations in the post-transplant setting for patients with RR AML (NCT3600155) and MDS (NCT02530463). Furthermore, nivolumab was studied in a front-line setting combined with idarubicine and cytarabine. There were 42 patients with AML enrolled; 50% had adverse ELN genetic risk and 18% *TP53* mutations [144]. The combination led to an ORR of 80% including 64% CR and 14% CRi/CRp with a median OS of the whole cohort being 18.5 months and for those who proceeded to allo-HCT being 25 months. Finally, a phase II pilot study assessed the role of nivolumab as maintenance therapy in high-risk AML showing a modest ability to extend remissions, providing no support for use as a single agent in the post-HCT setting [145].

**Pembrolizumab:** Pembrolizumab is a monoclonal antibody targeting the anti–programmed death-1 (anti-PD1) protein found on T-cells. The combination of pembrolizumab + AZA was studied in a multicentric phase II study [146] in 37 patients with newly diagnosed and relapsed refractory AML aged >65, and 29 of the 37 patients were evaluable for response with an ORR of 55% (CR/CRI: 14%, PR: 4%, hematological improvement: 14%, stable 24%) with a median OS of 10.8 months. Seventeen of twenty-two patients with newly diagnosed AML were evaluable for response with an ORR of 94% (CR/Cri: 47%) with a median OS of 13 months [146]. The combination was well tolerated without major toxicities, with better efficacy in the first-line setting. A smaller study investigated the role of [147] decitabine + pembrolizumab in 10 patients with relapsed AML. ORR was observed in six patients with a median OS of 10 months. Zeidner et al. [148] conducted a phase II study in 37 patients with relapsed–refractory AML treated with high-dose cytarabine + pembrolizumab. The ORR was 46% (Cr/cri: 38%) with a median OS of 11.1 months. The greatest benefit was observed in patients with the treatment as a first salvage regimen. Patients with ASXL1 mutations achieved a better ORR (50%), and two of five patients enrolled with *TP53* mutations achieved CRc. A retrospective analysis [149] investigated the potential benefit of the use of pembrolizumab prior to allo-HCT. The results did not show a benefit in terms of OS and RFS, and no increase in grade III-IV acute graft-versus-host disease was seen in those who received ICI prior to allo-HCT compared with historical controls. To date, there are many trials that will better elucidate the role of pembrolizumab-based combinations in the setting of newly diagnosed and relapsed AML combined with HMA + VEN (NCT03969446; NCT04284787) and to eradicate MRD pretransplant combined with chemotherapy (NCT04214249). Pembrolizumab and AZA has also been studied in high-risk MDS, showing no benefit in patients with high-risk MDS after the failure of HMA agents. For 17 patients not pretreated with HMA, the ORR was 76% (CR: 18%), whereas in the cohort of patients pretreated with HMA, the ORR was only 25% (CR: 5%) [150].

### 8.3. Poly(ADP-Ribose) Polymerase (PARP)-Inhibitor-Based Combinations

Talazoparib has been studied in early phase I-II clinical trials for AML as a monotherapy, revealing limited efficacy (NCT01399840) [151]. Better results are expected in cohesin mutant AML (NCT03974217) characterized by mutations in genes such as STAG2, SMC1A, RAD21, PDS5B, and SMC3, as previously described. Preclinical research indicates that combining talazoparib with decitabine, a DNA demethylating agent, enhances PARP1 recruitment and inhibits DNA repair, leading to synergistic cytotoxicity in AML cells [152]. A phase I clinical trial reported the results of decitabine combined with talozoparib in relapse–refractory AML [153]. Responses included complete remission with incomplete count recovery observed in 2 patients (8%) of 24 and hematologic improvement in 3. The combination was well tolerated. Furthermore, talazoparib is being investigated in combination with gemtuzumab ozogamicin (GO), an anti-CD33 antibody conjugated to calicheamicin, recently approved by the FDA for treating CD33-positive AML (NCT04207190) [154]. Despite the lack of robust data supporting the use of PARP inhibitors in AML, there is potential for successful treatment, particularly in cohesin mutant AML and through combination therapies involving agents like decitabine. As previously discussed, *STAG2*-mutated AML can be more sensitive to immune checkpoint inhibition, in particular to anti-PD(L)1 immunotherapy. The efficacy of combinatorial approaches including PARPi and ICB remains to be assessed in this specific setting.

### 8.4. Regimens Including Menin Inhibitors for KMT2A-Mutated AML

The phase I/II AUGMENT-101 trial (NCT04065399) is currently assessing the efficacy of revumenib monotherapy in adult and pediatric patients with relapsed or refractory acute leukemia characterized by a KMT2A rearrangement or NPM1 mutation. Recently updated findings from this trial were presented at the ASH meeting 2023 [155], where 94 patients were enrolled, with a median age of 37 years. These patients had undergone extensive prior treatments, with a median of two prior lines of therapy. With a median follow-up of 6.1 months in the efficacy population, the overall response rate was found to be 63%, with 23% of patients achieving complete remission or complete remission with partial hematologic recovery. Moreover, recognizing the heightened susceptibility of KMT2A-rearranged (KMT2Ar) leukemias to apoptosis induction through BCL2 inhibition, recent observations have shown synergistic activity in models of KMT2Ar or NPM1-mutated (NPM1mt) leukemia with dual Bcl-2 and menin inhibition [156]. As a result, the phase I/II SAVE trial (NCT05360160) is investigating the combination of revumenib with venetoclax and the hypomethylating agent ASTX727, showing promising results. Further expanding on this approach, another study (NCT06177067) is evaluating the combination of revumenib with VEN + AZA in front-line AML patients to assess both the safety and efficacy profiles of this triplet regimen. These collective findings underscore the potential significance of menin inhibitors as crucial therapeutic targets for patients with *KMT2A*-mutated acute leukemia, with ongoing evaluation of combinatorial strategies offering promising avenues for further exploration and potential clinical benefit.

### 8.5. Combinatorial Strategies Targeting the Interleukin 3 Receptor CD123

CD123 is a subunit of the interleukin 3 (IL-3) receptor expressed on the surface of blasts in most AML and, in particular, in poor-risk genetic subgroups and high cell count at diagnosis (Figure 1) [157]. Tagraxofusp (sl-401) (TAG) is a recombinant protein drug targeting CD123 and is currently approved as monotherapy for the treatment of blastic plasmacytoid dendritic cell neoplasm (BPDCN).

In a phase Ib trial (NCT03113643), the combination of TAG + AZA and VEN showed promising results in AML, MDS, and BPDCN, with 89% of patients achieving complete responses. This activity was observed across all genetic subgroups, including TP53-mutated AML/MDS and secondary AML. An expansion cohort in newly diagnosed AML, reported by Lane et al. [158], treated 26 adverse-risk patients according to the ELN 2022 criteria, with 50% having TP53 mutations. Of these, 39% achieved complete remission (CR), with an additional 19% achieving incomplete CR, and a median OS of 14 months in the overall population, reduced to 9.5 months in the TP53-mutated subgroup. Ongoing trials, such as NCT05442216, are investigating the role of TAG in combination with AZA ± VEN specifically in secondary AML. Moreover, it has been studied as a single agent for consolidation therapy in AML patients at high risk of relapse and who are MRD positive (NCT02270463).

Pivekimab Sunirine (IMGN632) (PVEK) is an antibody-drug conjugate (ADC) consisting of a high-affinity CD123 antibody, a cleavable linker, and an indolinobenzodiazepine pseudodimer (IGN) payload. The IGN payload induces DNA alkylation and single-strand breaks without crosslinking, demonstrating high potency against tumor cells while exhibiting reduced toxicity to normal marrow progenitors compared to other DNA-targeting payloads. Preliminary clinical data for relapsed–refractory AML (R/R AML) [159] support the ongoing investigation of the PVEK + AZA + VEN triplet combination therapy (NCT04086264).

Flotetuzumab (MGD006), a bispecific antibody engineered to bind both CD3 and CD123 on AML cells, is currently undergoing investigation in a phase I/II trial (NCT02152956) for R/R AML [84]. Among the 88 patients enrolled in the trial, the ORR was reported as 13.6%, with 11.7% achieving CR. Across all dosing cohorts, a reduction in BM blasts has been observed, indicating the potential efficacy of the treatment.

These findings suggest that anti-CD123-directed therapies (Figure 1) hold promise as a therapeutic option for patients with R/R AML and high-risk genetic profiles, demonstrating activity in reducing leukemic cell burden and achieving complete remission in a subset of patients. Further investigation through ongoing clinical trials will provide additional insights into its safety and efficacy profile, potentially leading to improved outcomes for AML patients.

## 9. Conclusions and Perspectives

AML with adverse genetics encompasses a complex heterogeneous disease driven by different genetic abnormalities and mostly associated with an immune-hot microenvironment, with high inflammation scores, an increased T-cell infiltration, and upregulation of co-inhibitory receptors such as PD-L1. However, early-phase trials with ICB or monoclonal antibodies as monotherapy strategies have shown only modest or no impact on patients’ outcomes in this AML category, indicating the urgent need to provide strong biological rationales for combinatorial approaches, such as those with small molecules. In this regard, encouraging evidence has been provided from preclinical studies with APR-246/eprenatopopt or the BCL2 inhibitor/venetoclax, which, beyond inducing cytotoxicity against AML cells, can both enhance immune responses against leukemia. Combinatorial approaches should also be investigated along with other immunotherapeutic modalities, including T-cell engagers or AML vaccines. In this regard, preclinical evidence has shown potential synergism between AMG-330 BiTE, which is being investigated in phase 1/2 trials, and ICB to enhance T-cell proliferation and cytotoxicity against leukemia cells. Further, the design of next-generation vaccines will benefit from large-scale proteomic investigation of the AML surfaceome, aimed at identifying novel AML-restricted antigens with high immunogenicity. Anti-leukemic vaccination should be rationally used during AML evolution, as it could benefit high-risk preleukemic states to prevent disease progression or can eradicate MRD persistence after allogeneic transplantation. Novel therapeutic modalities including metal-based or bioinspired nanodrugs, as well as CRISPR-based platforms targeting oncogenic drivers or leukemic stem cells should be thoroughly investigated in AML with adverse genetics in combination with immunotherapies. These new treatment approaches can effectively enhance the delivery of biologics/immunomodulators with tailored (temporally and spatially defined) reactivation of the immune system against leukemia.

## Figures and Tables

**Figure 1 ijms-25-04259-f001:**
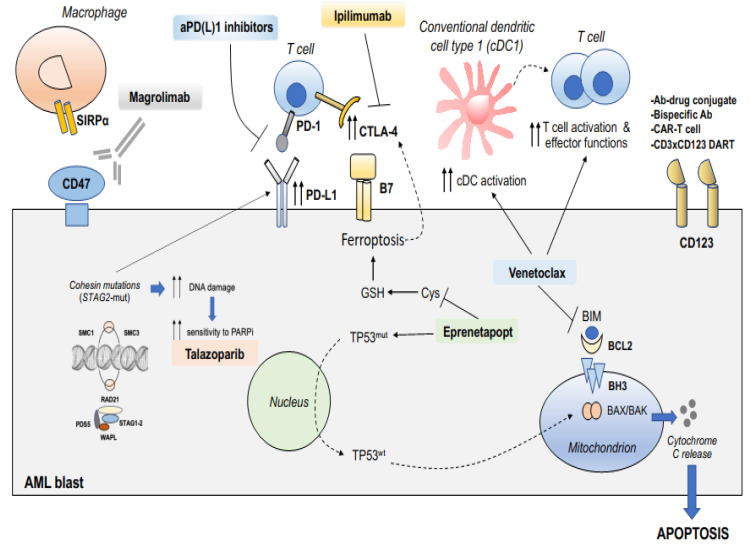
Biological rationales for combining targeted therapies and immunotherapies in AML with adverse genetics. Displayed are multiple signaling pathways that are often activated in high-risk AML and can be targeted by a combination of small molecule drugs and immunotherapies. The BCL2 inhibitor venetoclax effectively induces mitochondrial apoptosis in leukemic cells and activates conventional dendritic cells to enhance anti-tumor T-cell immunity. APR246/eprenetapopt re-establishes wild-type TP53 tumor suppressor function in TP53mut AML, inducing apoptosis and upregulating co-inhibitory molecules such as CTLA-4 within the leukemic microenvironment, thereby increasing sensitivity to immune checkpoint blockade (ICB) therapy. In cohesin-mutated AML, particularly in cells with STAG2 mutations, DNA repair and replication pathways are identified as genetic vulnerabilities. Consequently, STAG2mut AML cells exhibit increased sensitivity to poly(ADP-ribose) polymerase (PARP) inhibition. Cohesin-deficient leukemic cells also demonstrate elevated expression of the PD-L1 immune checkpoint molecule, which can be targeted by ICB therapy, as with anti-PD(L)1 inhibitors. Another strategy against HR-AML is represented by targeting the CD123 molecule, which is found to be overexpressed in AML. CD123 expression can be further enhanced by hypomethylating agents, such as azacytidine (not shown).

**Table 1 ijms-25-04259-t001:** High-risk genetic features at diagnosis in AML adapted from ELN 2022 [1].

High-Risk Genetic Features
t(6;9)(p23.3;q34.1)/DEK::NUP214 t(v;11q23.3)*KMT2A*-rearranged t(9;22)(q34.1;q11.2)/*BCR::ABL1* (BCR-ABL+) t(8;16)(p11.2;p13.3)/*KAT6A::CREBBP* inv(3)(q21.3q26.2) or t(3;3)(q21.3;q26.2) *GATA2*, *MECOM(EVI1)* t(3q26.2;v)/*MECOM*(*EVI1*)-rearranged −5 or del(5q); −7; −17/abn(17p) Complex karyotype (CK) Monosomal karyotype (MK) Mutated RUNX1 Mutated EZH2 Mutated ASXL1 Mutated BCOR Spliceosome mutations (SRSF2, SF3B1, U2AF1, ZRSR2) Mutated STAG2 Mutated TP53

**Abbreviations:** t: translocation; (): grouping for breakpoints and structurally altered chromosomes; ;: separates rearranged chromosomes and breakpoints involving more than one chromosome; p: short arm of a chromosome; q: long arm of a chromosome; ::: break and join; inv: inversion; −: loss of a chromosome; abn: abnormalities.

**Table 2 ijms-25-04259-t002:** Clinical trials combining immunotherapy and target therapy in HR genetic AML.

Drug	Combinations	Mutation	Clinical Trials	Disease		Outcomes
APR-246 small molecule that targets TP53-mutated cancers	APR-246 + AZA APR-246 + AZA APR-246 + VEN + AZA	TP53 TP53 TP53	NCT03072043 (PHASE IB-II) NCT03588078 (PHASE II) NCT04214860 (PHASE I)	40 MDS 11 AML 34 MDS 18 AML 49 AML	1 1 1	ORR 71%, CR 44 Median OS 10.8 mo ORR 52% CR 37% Median OS 12.1 mo ORR 64% CR 38%
MAGROLIMAB monoclonal antibody against CD47 and macrophage checkpoint inhibitor	MAGROLIMAB + AZA MAGROLIMAB + AZA vs. VEN-AZA or chemo MAGROLIMAB + AZA-VEN vs. placebo + AZA + VEN	TP53 TP53 TP53	NCT03248479 (PHASE I) NCT04778397 (PHASE III) NCT05079230 (PHASE III)	87 AML (82.8% TP53) Ongoing Ongoing	1 1 1	ORR 47.2% CR 31.9% median OS 9.8 mo Ongoing Ongoing
SABATOLIMAB Checkpoint inhibitor anti TIM3 monoclonal antibody	SABATOLIMAB + HMA SABATOLIMAB + AZA + VEN	All;HR AML All	NCT03066648 (PHASE Ib) NCT04150029 (PHASE II)	53 MDS 48 AML Ongoing	1 1	ORR AML 40% CR30%; HR AML ORR 53% median duration of response 12 months Ongoing
NIVOLUMAB Checkpoint inhibitor antiPD-1 monoclonal antibody approved for different types of cancers	NIVOLUMAB + AZA NIVOLUMAB + AZA + IPILIMUMAB NIVOLUMAB + CHEMO	All All All (50% HR)	NCT02397720 (PHASE II) NCT02397720 (PHASE II) NCT02464657 (PHASE II)	70 AML 31 AML 42 AML	>1 >1 1	ORR 33%, CR22% median OS 6.2 mo (ASLX1 better response) ORR 46%, CR36% median OS 10.5 mo ORR 80%, CR64% median OS 18.5 mo
PEMBROLIZUMAB Checkpoint inhibitor antiPD-1 monoclonal antibody approved for different types of cancers	PEMBRO + AZA PEMBRO + ARA C PEMBRO + DEC +/− VEN PEMBRO + AZA + VEN PEMBRO + CHEMO	All All All All All	NCT02845297 (PHASE II) NCT02768792 (PHASE II) NCT03969446 (PHASE II) NCT04284787 (PHASE II) NCT04214249 (PHASE II)	37 AML(17 first line) 37 AML Ongoing Ongoing Ongoing	≥1 >1 ≥1 ≥1 1	ORR 55%, CR14% median OS 10.8 mo newly diagnosed ORR94%, CR47% median OS 13 mo ORR46%, CR38% median OS 11 mo (ASLX1 better response) Ongoing Ongoing
TALAZOPARIB PARP inhibitor approved for breast cancer	TALAZOPARIB + DEC TALAZOPARIB BASED TALAZOPARIB + GO	All Cohesin mutated Cd33+	NCT02878785 (PHASE I) NCT03974217 (PHASE I) NCT04207190 (PHASE I)	24 AML Ongoing Ongoing	>1 ≥1 >1	CR 8% Ongoing Ongoing
REVUNEMIB Menin inhibitor FDA approved in adult and pediatric relapsed or refractory (R/R) KMT2A-rearranged acute leukemia	REVUNEMIB + VEN + ASX727 REVUNEMIB + VEN + AZA	All All	NCT05360160 (PHASE II) NCT06177067 (PHASE II)	Ongoing Ongoing	1 >1	Ongoing Ongoing
TAGRAXOFUSP CD123-directed cytotoxin approved as monotherapy for the treatment of blastic plasmacytoid dendritic cell neoplasm PIVEKIMAB Antibody drug conjugate targeting CD-123	TAGRAXOFUSP + AZA + VEN PIVEKIMAB + AZA + VEN	HR AML Cd123+	NCT03113643 (PHASE IB) NCT04086264 (PHASE IB-II)	Ongoing (preliminary results 26 AML HR) Ongoing	1 ≥1	Ongoing preliminary results CR 39% median OS 14 mo; median OS TP53 9.5 mo Ongoing

**Abbreviations:** APR-246: eprenetapopt; AZA: azacytidine; VEN: venetoclax, MDS: myelodysplastic syndrome; AML: acute myeloid leukemia; HR: high risk, ORR: overall response rate, CR: complete remission; OS: overall survival, PEMBRO: pembrolizumab; DEC: decytabine; GO: gentuzumab.

## Data Availability

Not applicable.

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
