# Peer review of "Precision Medicine Approaches in Acute Myeloid Leukemia with Adverse Genetics"

_ijms, 2024, doi:10.3390/ijms25084259_

Round 1

Reviewer 1 Report

Comments and Suggestions for Authors

The authors reported the recent advances of HR AML from the description of the HR genetic features to the target therapy developed. 

1) please revise English, some editing is needed across the manuscript

2) Form line 275 the statement is not really clear, please rephrase it

3) Table 2 is really interesting and well done. I would suggest to include another table with a resuming name of the drug, where it acts, if in clinical trial (phase) or already approved. This could help the reader

Comments on the Quality of English Language

Please revise some sentences where minor English revision is required. Ex: line 25 " accounts for " in this context maybe accounting for would have been more appropriate. 

Author Response

The authors reported the recent advances of HR AML from the description of the HR genetic features to the target therapy developed.  1) please revise English, some editing is needed across the manuscript; 2) Form line 275 the statement is not really clear, please rephrase it; 3) Table 2 is really interesting and well done. I would suggest to include another table with a resuming name of the drug, where it acts, if in clinical trial (phase) or already approved. This could help the reader

We thank the reviewer for the comment. We revised English across the manuscript and rephrased line 275, accordingly. We added a column on table 2 describing the mechanism of actions of the principal drugs discussed and if already approved. Whereas the main focus of the paper is to discuss combination strategies in HR AML, we reported in the table the clinical trials involving these strategies.

Comments on the Quality of English Language. Please revise some sentences where minor English revision is required. Ex: line 25 " accounts for " in this context maybe accounting for would have been more appropriate. 

We revised some sentences across the manuscript to improve English, as it is the case of line 25.

Reviewer 2 Report

Comments and Suggestions for Authors

The review “Precision medicine approaches in acute myeloid leukemia with adverse genetics” is devoted to the novel approaches for the treatment of acute myeloid leukemia with adverse genetics. The main question addressed by the research, is the consideration of specific mutations that occur in acute myeloid leukemia and ways of treating the disease in each specific case. Each of the mutations, the genomic regions and proteins involved is discussed in detail, modern treatment methods and combinations of methods (immunotherapy/targeted therapy) are provided.

The work provides an in-depth analysis of modern literary sources. The authors have reworked the material so that it is understandable for researchers. Original, well-structured tables are presented (not simply copying sources), which provide a clear understanding of high-risk genetic features (Table 1) and possible therapy (Table 2) for myeloid leukemia with adverse genetics. This presentation of material is important for researchers in many fields: doctors, biochemists etc

However, it is worth noting that this review is based on the results of other groups, since from this group of researchers only the last author has work on the topic. If this is not the case, it is worth adding works on this topic.

As for the description of methods, the authors describe the drugs in quite detail, but nothing is said about the various approaches to immune therapy. It is worth adding to the review the possibilities of specific immunotherapy approaches, namely, the bispecific T-cell engager platform, AMG330. What experimental methods are currently being tested in clinical trials? The work completely lacks information on the possible use of vaccines. It's worth adding this information to the review.

The review would also benefit from considering the possibility of using bioinspired nanomaterials in the field of study.

Conclusions and references are mostly relevant. The issues addressed in the review have also been largely resolved.

Tables and figures are of high quality. However, it should be noted that the specific designations are not understandable to a wide class of researchers (for example, designations for inversions, translocations etc. in Table 1). Authors are strongly encouraged to include an abbreviations section in their review, adding abbreviations from the text and tables.

Minor:

Lines 29, 550-573. Non-standard font.

Line 47. It is advisable to add a link to reference 1.

Line 351. Typo: phagocvtosis à phagocytosis

Author Response

The review “Precision medicine approaches in acute myeloid leukemia with adverse genetics” is devoted to the novel approaches for the treatment of acute myeloid leukemia with adverse genetics. The main question addressed by the research, is the consideration of specific mutations that occur in acute myeloid leukemia and ways of treating the disease in each specific case. Each of the mutations, the genomic regions and proteins involved is discussed in detail, modern treatment methods and combinations of methods (immunotherapy/targeted therapy) are provided. The work provides an in-depth analysis of modern literary sources. The authors have reworked the material so that it is understandable for researchers. Original, well-structured tables are presented (not simply copying sources), which provide a clear understanding of high-risk genetic features (Table 1) and possible therapy (Table 2) for myeloid leukemia with adverse genetics. This presentation of material is important for researchers in many fields: doctors, biochemists etc However, it is worth noting that this review is based on the results of other groups, since from this group of researchers only the last author has work on the topic. If this is not the case, it is worth adding works on this topic. As for the description of methods, the authors describe the drugs in quite detail, but nothing is said about the various approaches to immune therapy. It is worth adding to the review the possibilities of specific immunotherapy approaches, namely, the bispecific T-cell engager platform, AMG330. What experimental methods are currently being tested in clinical trials? The work completely lacks information on the possible use of vaccines. It's worth adding this information to the review. The review would also benefit from considering the possibility of using bioinspired nanomaterials in the field of study.

We thank the reviewer for all these comments. Specifically, in the revised version of the manuscript, we discuss these other immunotherapeutic approaches, including BiTE, AML vaccination, and bio-inspired nanomaterials (please refer to lines 278-330, and lines 541-550, of the revised version of the manuscript with tracked changes). We have particularly focused the attention on potential synergism with other immune-based drugs and on the correct timing of their usage during AML evolution, which however should be tested within ad-hoc clinical trials in the near future.

Conclusions and references are mostly relevant. The issues addressed in the review have also been largely resolved. Tables and figures are of high quality. However, it should be noted that the specific designations are not understandable to a wide class of researchers (for example, designations for inversions, translocations etc. in Table 1). Authors are strongly encouraged to include an abbreviations section in their review, adding abbreviations from the text and tables

We thank for this comment. We have added the abbreviations section in Table 1 and revised all abbreviations in the text and in all tables.

Minor:

Lines 29, 550-573. Non-standard font.

Line 47. It is advisable to add a link to reference 1.

Line 351. Typo: phagocvtosis à phagocytosis

All these minor revisions have been corrected in the revised manuscript.

Reviewer 3 Report

Comments and Suggestions for Authors

Even though the present article has selected and researched suitable objectives for study and discussion, the following corrections seem necessary before publishing the article.
1- The introduction section is very short and there is no mention of new approaches to cancer treatment such as scaffolds containing drugs or anti-cancer ions or the use of enzymes in targeted treatment. Please read and use the following.

https://link.springer.com/article/10.1007/s10924-022-02615-x

https://www.sciencedirect.com/science/article/pii/S0753332223010831

2- In the whole study, the use of drugs based on monoclonal antibodies has been investigated. Shouldn't new methods such as CRISPR be examined or mentioned in this study?

3- The conclusion and perspective part is written very vaguely. Please rewrite it.

Author Response

Even though the present article has selected and researched suitable objectives for study and discussion, the following corrections seem necessary before publishing the article.
1- The introduction section is very short and there is no mention of new approaches to cancer treatment such as scaffolds containing drugs or anti-cancer ions or the use of enzymes in targeted treatment. Please read and use the following.

https://link.springer.com/article/10.1007/s10924-022-02615-x

https://www.sciencedirect.com/science/article/pii/S0753332223010831

We thank the reviewer for their comment. We acknowledge that the original version of the manuscript did not mention new cancer treatments, but we have included some insights into these in the revised version (please refer to lines 541-549 of the revised manuscript with tracked changes). In this context, we have cited published articles about new approaches using bio-inspired nanomaterials and metal-ion drugs for next-generation immunotherapy (references 131 and 133 of the revised manuscript), as well as a congress communication about the use of nanomaterials specifically in AML.

We strongly disagree with the reviewer's suggestion to use the previously mentioned articles, as they are not relevant to our field of research, which focuses on AML with adverse genetics. Indeed, the first article (https://link.springer.com/article/10.1007/s10924-022-02615-x) pertains to bone tissue engineering, and the second (https://www.sciencedirect.com/science/article/pii/S0753332223010831) primarily discusses improving methotrexate toxicity with glucarpidase. Particularly, the latter paper presents no evidence that would suggest any rationale for a synergistic effect of glucarpidase with immunotherapy in high-risk AML

2- In the whole study, the use of drugs based on monoclonal antibodies has been investigated. Shouldn't new methods such as CRISPR be examined or mentioned in this study? We thank the reviewer for this comment. Accordingly, we add a brief sentence on the usage of CRISPR-based therapeutics in the field of AML (lines 549-550). Indeed, very little has been published in this field, and no study has investigated CRISPR-based drugs in HR-AML.

3- The conclusion and perspective part is written very vaguely. Please rewrite it. We follow the reviewer suggestion, and rewrite the conclusion of our review article (please refer to lines 760-786 of the revised version of the manuscript with tracked changes).

Reviewer 4 Report

Comments and Suggestions for Authors

In this submitted paper, the authors tried to discuss the recent developments in uncovering the contribution of cell-intrinsic mechanisms and the immune system in the leukemogenesis of myeloid cells causing acute myeloid leukemia (AML). They have mainly focused on the biological foundations for integrating the approaches of targeted therapy and immunotherapy, which are presently being examined in current trials and could optimistically improve the poor outcomes for AML patients. The subject of the paper is interesting along with an acceptable writing language and the latest achievements are all tried to summarize here. The only thing could be the conclusion section which needs to be appropriately expanded.

Author Response

In this submitted paper, the authors tried to discuss the recent developments in uncovering the contribution of cell-intrinsic mechanisms and the immune system in the leukemogenesis of myeloid cells causing acute myeloid leukemia (AML). They have mainly focused on the biological foundations for integrating the approaches of targeted therapy and immunotherapy, which are presently being examined in current trials and could optimistically improve the poor outcomes for AML patients. The subject of the paper is interesting along with an acceptable writing language and the latest achievements are all tried to summarize here. The only thing could be the conclusion section which needs to be appropriately expanded.

We thank the reviewer for the nice comment. Accordingly, we have appropriately expanded the conclusion section of our review article (please refer to lines 760-786 of the revised version of the manuscript with tracked changes).